# TestRank: Bringing Order into Unlabeled Test Instances for Deep Learning Tasks

**Yu Li[†], Min Li[†], Qiuxia Lai[†], Yannan Liu[\*], and Qiang Xu[†]**
[†] CURE Lab, The Chinese University of Hong Kong
[\*] Wuheng Lab, ByteDance
{yuli,mli,qxlai,qxu}@cse.cuhk.edu.hk

## Abstract

Deep learning (DL) systems are notoriously difficult to test and debug due to the lack of correctness proof and the huge test input space to cover. Given the ubiquitous unlabeled test data and high labeling cost, in this paper, we propose a novel test input prioritization technique, namely *TestRank*, which aims at revealing more model failures with less labeling effort. TestRank brings order into the unlabeled test data according to their likelihood of being a failure, i.e., their *failure-revealing capabilities*. Different from existing solutions, TestRank leverages both *intrinsic* and *contextual* attributes of the unlabeled test data when prioritizing them. To be specific, we first build a similarity graph on both unlabeled test samples and labeled samples (e.g., training or previously labeled test samples). Then, we conduct graph-based semi-supervised learning to extract contextual features from the correctness of similar labeled samples. For a particular test instance, the contextual features extracted with the graph neural network and the intrinsic features obtained with the DL model itself are combined to predict its failure-revealing capability. Finally, TestRank prioritizes unlabeled test inputs in descending order of the above probability value. We evaluate TestRank on three popular image classification datasets, and results show that TestRank significantly outperforms existing test input prioritization techniques. Our code is available at: https://github.com/cure-lab/TestRank.

## 1 Introduction

Deep learning (DL) systems are prone to errors due to many factors, such as the biased training/validation dataset, the limitations of the model architecture, and the constraints on training cost. It is essential to conduct high-quality testing before DL models are deployed in the field; otherwise, the behaviors of DL models can be unpredictable and result in severe accidents after deployment. However, the cost of building test oracles (i.e., the ground-truth output) by manually labeling a massive set of test instances is prohibitive, especially for tasks requiring experts for accurate labeling, such as medical images and malware executables.

To tackle the above problem, various test input prioritization techniques [Feng et al., 2020, Byun et al., 2019, Shen et al., 2020] are proposed to identify 'high-quality' test instances from a large amount of unlabeled data, which facilitates revealing more failures (e.g., misclassification) of the DL model with reasonable labeling effort. These methods try to derive the failure-revealing capability of a test instance with its *intrinsic attributes* extracted from the responses of the model under test (e.g., the softmax-based probabilities given by the target DL model to this specific input). DeepGini [Feng et al., 2020] feeds the unlabeled data to the target DL model and calculates confidence-related scores based on the model's output probabilities to rank the unlabeled test cases. Test cases with nearly equal probabilities on all output classes are regarded as less confident ones and are likely to reveal

35th Conference on Neural Information Processing Systems (NeurIPS 2021).

model failures. Similarly, [Byun et al., 2019] use the uncertainty score obtained from MC-Dropout for test input prioritization. Multiple-boundary clustering and prioritization (MCP) [Shen et al., 2020] considers both the output probabilities and the balance among each classification boundary when selecting test cases. All existing works try to identify instances near the decision boundary and prioritize them. However, we argue that near-boundary instances are not necessarily failures, especially for well-trained classifiers with high accuracy. Also, as failures can be far from the decision boundary, existing methods could fail to reveal these remote failures.

To estimate a test instance's capability in revealing failures, in addition to the intrinsic attributes mentioned above, there is another type of information: the known classification correctness of labeled samples (i.e., training samples and previously tested samples) and their relationship to the unlabeled instance. This information provides extra insight into the target model's behavior. Such data is already known, and it provides *contextual information* that reflects the corresponding inference behaviors of the target model for a set of similar instances.

This work presents a novel test input prioritization technique, namely *TestRank*, for DL classifiers. TestRank exploits both intrinsic and contextual attributes of test instances to evaluate their failure-revealing capabilities. Based on the intuition that similar inputs are usually associated with the same classification results, we propose to use graph neural networks (GNNs) [Kipf and Welling, 2017] to summarize the neighboring classification correctness for each unlabeled instance into contextual attributes. GNNs have been well-studied and valued for their relational inductive bias for extracting graph information. Our method, TestRank, constructs a similarity graph on both unlabeled and labeled instances and apply the semi-supervised GNN learning to extract the contextual attributes. After that, we aggregate intrinsic (such attributes are extracted from the input samples without considering their neighbors) and contextual attributes with a neural-network-based binary classifier for test input prioritization.

The contributions of our work are as follows:

- To the best of our knowledge, *TestRank* is the first work that takes the contextual information from the target DL model into consideration for test input prioritization.

- We propose constructing a similarity graph on both labeled and unlabeled samples and training a graph neural network to extract useful contextual attributes from the contextual information for these unlabeled instances. We also present approximation techniques to reduce its computational complexity with minor impact on the performance of *TestRank*.

- We propose a simple yet effective neural network that combines the intrinsic attributes and contextual attributes of unlabeled test instances for their failure-revealing capability estimation.

We empirically evaluate *TestRank* on three popular image classification benchmarks: CIFAR-10, SVHN, and STL10. The results show that our method outperforms the state-of-the-art methods by a considerable margin.

## 2   Test Input Prioritization

Let us use $f : \mathcal{X} \rightarrow \mathcal{Y}$ to represent the given target DL model, where $\mathcal{X}$ and $\mathcal{Y}$ are the input and output space, respectively. For effective testing[1], the debugging center needs to perform test input prioritization, i.e., select a certain number of test instances from the large unlabeled test instance pool that can reveal as many failures as possible. Later, these failures are fed back to the training center for failure analysis and model repair. We define the model failures as follows:

**Definition 1.** ***DL Model Failure.*** *A failure of the DL model can be uncovered by the test instance* **x** *if the predicted label* $f(\mathbf{x})$ *is inconsistent with its ground truth label* $y_\mathbf{x}$, *namely* $f(\mathbf{x}) \neq y_\mathbf{x}$.

Formally, the debugging center selects and labels $b$ test cases $\mathbf{X}_S$ ($|\mathbf{X}_S| = b$) from the unlabeled test instance pool $\mathbf{X}_U$. The objective of test input prioritization is to maximize the detected failures:

$$\max |\{\mathbf{x} | f(\mathbf{x}) \neq y_\mathbf{x}\}|, \text{ where } \mathbf{x} \in \mathbf{X}_S \text{ and } |\mathbf{X}_S| = b. \tag{1}$$

---

[1]Please note that, we focus on testing the functional correctness of the DL model, and we assume the collected testing data are clean samples instead of maliciously generated ones.

Different solutions are proposed to quantify the failure-revealing capability of unlabeled instances. DeepGini [Feng et al., 2020] proposes to evaluate a single test instance via the DL model's final statistical output: $f(t) = 1 - \Sigma_{i=1}^{N} p_{t,i}^2$, where $p_{t,i}$ is the predicted probability that the test case $t$ belongs to the class $i$. Given the sum of $p_{t,i}$ is 1, impurity function $f(t)$ is maximal when all $p_{t,i}$ values are equal. DeepGini also adopts the neuron coverage criteria proposed in DeepXplore [Pei et al., 2017] and DeepGauge [Ma et al., 2018] in test input prioritization, and the result shows that the impurity-based selection is much better than coverage-based selection.

Instead of evaluating the overall likelihood of failure for all classes, Multiple-Boundary Clustering and Prioritization (MCP) proposes to evaluate it for each pair of classes individually [Shen et al., 2020]. In this way, test instances can be evenly selected for each class pair and the failure cases are investigated at the finer granularity. Besides these metrics, [Byun et al., 2019] also propose to measure the likelihood of incorrect prediction by the uncertainty of the model's output, which reflects the degree to which a model is uncertain about its prediction. In practice, evaluating uncertainty requires the task DL model to be a Bayesian Neural Network [Richard and Lippmann, 1991, Neal, 2012] or containing a dropout layer for approximation [Gal and Ghahramani, 2016].

Besides examining the DL model's final outputs, [Kim et al., 2019] proposes two surprise adequacy (SA) criteria that make use of the target DL's internal outputs (e.g., the activation traces). They are Likelihood-based Surprise Adequacy Coverage (LSA) and Distance-based Surprise Adequacy Coverage (DSA). LSA and DSA measure the likelihood or distance of an unlabeled instance to the training instances, respectively. Test samples with higher SA values are preferred in testing.

To sum up, all existing methods use the target model's outputs to one input, i.e., its intrinsic attributes, for its failure-revealing capability estimation. In contrast, we make use of both intrinsic and contextual attributes of an instance for better estimation (see later sections for details).

## 3 TestRank

### 3.1 Motivation

The failure-revealing capability of an unlabeled test input is closely related to its attributes for the DL model under test. In this work, we distinguish two kinds of attributes for an unlabeled instance: the *intrinsic attributes* and the *contextual attributes*.

We define the *intrinsic attributes* of an input as the output responses assigned by the target DL model to this specific input. It could be, for example, the predictive output distribution of the input from the target DL model, reflecting the *sentiment* derived from the computation performed by the target model [Byun et al., 2019]. This kind of attributes is adopted by existing test input prioritization approaches [Feng et al., 2020, Shen et al., 2020, Byun et al., 2019]. Note that we define such attributes as 'intrinsic'

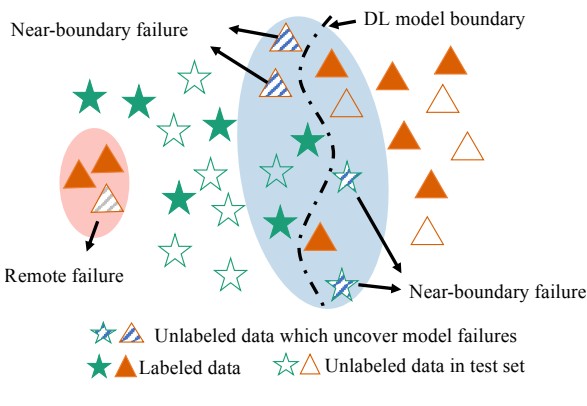

Figure 1: A motivational example.

because they are extracted from inputs without considering their context, *i.e.*, the classification correctness of its similar instances.

In contrast with the intrinsic attributes, the *contextual attributes* provide a deeper insight into the target model for the unlabeled samples: the contextual attributes for an unlabeled sample summarize the classification correctness of similar and labeled samples. For a particular test instance, such contextual attributes are useful and complementary to the intrinsic attributes.

An illustrative example is shown in Figure 1, wherein we visualize the behavior of a two-class classifier on the unlabeled test data and historically labeled data distribution. The blue region includes the instances that are near the decision boundary. Intuitively, the classifier is uncertain about the data when data is near the decision boundary and is likely to misclassify it. Existing

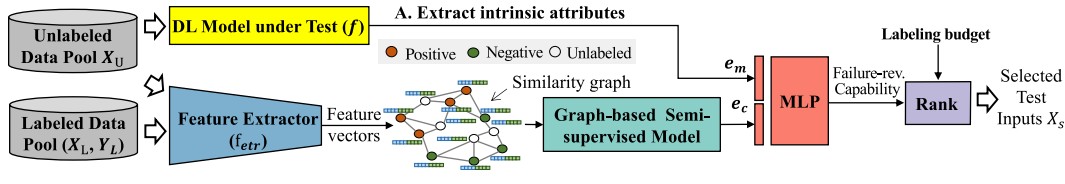

Figure 2: The overview of TestRank.

works [Shen et al., 2020, Feng et al., 2020, Byun et al., 2019] propose various indicators (*e.g.*, confidence/uncertainty/surprise scores) to help identify the near-boundary instances. However, the near-boundary instances are not necessarily failures, and some of them can be correctly classified by a well-trained classifier. What is worse, such testing approaches fail to capture the failures lying far from the decision boundary (*i.e.*, remote failures, shown in the red region in Figure 1), because DL models usually output high confidence (or low uncertainty) for these inputs. These failures may be caused by limited model capacity, insufficient training data, etc.

Our key insight is that we can use the contextual information (*e.g.* the classification correctness of similar labeled samples) to help locate both near-boundary and remote failures. The usefulness of the contextual information is due to the local continuity property [Bishop, 2006], which means that inputs close in the feature space share similar prediction behavior, e.g., classification results from the target model. As shown in Figure 1, some already labeled data, whose classification correctness is already known, surround the unlabeled data. If an unlabeled instance is close to already falsely classified data, under the local continuity property, it is likely that this instance is also a model failure. This property motivates us to extract the contextual attributes for an unlabeled instance from its neighboring labeled data. By combining the extracted contextual attributes with the intrinsic attributes, we expect to achieve better failure-revealing capability estimation.

## 3.2 Overview

Figure 2 shows the overview of TestRank, which consists of two attribute extraction paths for the final failure-revealing capability estimation:

1. **Path *A*: intrinsic attributes extraction**. Given a pool of unlabeled inputs $\mathbf{X}_U$, we use the target DL model $f$ to extract the intrinsic attributes $\mathbf{e}_m$ for each input. More precisely, we collect the output logits (i.e., vectors before the *softmax* layer) from the DL model as $\mathbf{e}_m$.

2. **Path *B*: contextual attributes extraction**. First, we use a feature extractor to map the original data space into a more compact feature space with good local continuity property. Then, we create a similarity graph (i.e., $k$-Nearest Neighbor Graph) based on the obtained feature vectors and their corresponding classification correctness, if any, from both unlabeled data pool $\mathbf{X}_U$ and labeled data pool $\mathbf{X}_L$ (e.g. training set and previously labeled test samples). Last but not least, the graph-based representation learning technique is applied to extract the contextual attributes $\mathbf{e}_c$ for each unlabeled instance. The details are elaborated in Sec. 3.3.

After attributes $\mathbf{e}_m$ and $\mathbf{e}_c$ are extracted, we combine them together via a Multi-Layer Perceptron (MLP) (see Sec. 3.4 for details). The MLP is responsible for predicting the failure-revealing ability for unlabeled test instances. At last, these instances are ranked according to their failure-revealing capability, and the top ones are selected under the given labeling budget.

As intrinsic attributes extraction is straightforward, we discuss the path *B* and how to combine path *A* and *B* in detail in the following subsections.

## 3.3 Contextual Attributes Extraction

We represent the contextual information from the DL model as a set of labeled inputs $\mathbf{X}_L$ and the corresponding classification correctness $\mathbf{Y}_L \in \{0, 1\}$, where correctly classified inputs are labeled as 0 and misclassified ones are labeled as 1. Given the contextual information, our goal is to extract the contextual attributes for each unlabeled test instance $\mathbf{x} \in \mathbf{X}_U$. However, extracting contextual attributes from labeled and unlabeled data is a non-trivial task because of the following reasons.

---

**Algorithm 1:** GNN-based Contextual Attributes Extraction

---

**Input:** Input samples $\mathbf{X}_U \cup \mathbf{X}_L$, Correctness of labeled samples $\mathbf{Y}_L$, Number of neighbors $k$, Feature extractor $f_{etr}$, number of GNN layers $M$, number of training epochs.

**Output:** Contextual attributes $\mathbf{E}_c$ for $\mathbf{X}_U$

1   $\overline{\mathbf{X}} = f_{etr}(\mathbf{X}_U \cup \mathbf{X}_L)$    # Extract compact representation;

2   $\mathbf{Edge} = knn\_graph(\overline{\mathbf{X}}, k)$.    # KNN Graph construction;

3   $\tilde{\mathbf{A}} = \mathbf{Edge} + \mathbf{I}_N$, $\tilde{\mathbf{D}} = \sum_j \tilde{\mathbf{A}}_{i,j}$;

4   $\mathbf{H}^0 = \overline{\mathbf{X}}$;

5   **for** *number of training epochs* **do**

6     **for** $l = 0, 1, \ldots, M-1$ **do**

7       $\mathbf{H}^{l+1} = \sigma(\tilde{\mathbf{D}}^{-\frac{1}{2}} \tilde{\mathbf{A}} \tilde{\mathbf{D}}^{-\frac{1}{2}} \mathbf{H}^l \Theta^l)$,

8     **end**

9     $loss = \text{CrossEntropyLoss}(\mathbf{H}^M, \mathbf{Y}_L)$;

10     Update $\Theta$;

11   **end**

12   $\mathbf{E}_c = \mathbf{H}^{M-1}[\text{index of } \mathbf{X_u}]$    # extract the representation from the $M-1_{th}$ GNN layer;

13   **return** $\mathbf{E}_c$;

---

First, it is well-known that the real data, especially image data, usually locates in high-dimensional space, wherein the underlying data distribution will live on complex and non-linear manifold embedded within the high-dimensional space [Bishop, 2006]. Therefore, constructing the relationships between different instances is difficult. To address the challenge, we adopt the representation learning process [Grill et al., 2020, van den Oord et al., 2018, Chen et al., 2020, He et al., 2020], which map the raw data into a compact feature space with better local continuity property, such that inputs close in the feature space share similar classification results. Thus, in the feature space, the proximity between inputs can be measured with simple distance metrics (*e.g.*, $L_2$, cosine).

Second, manually designing protocols to summarize the neighboring classification results is subject to the imperfection of local continuity. Namely, the result is easily affected by the noisy data in the neighboring space. To solve this challenge, we construct a similarity graph based on the labeled and unlabeled data, and then apply the more powerful graph learning technique – graph neural networks (GNN) – for contextual attribute extraction.

The GNN empowers graph embedding learning, as it employs a learnable aggregation and transform procedure [Kipf and Welling, 2017], which exploits the relational inductive-bias that exhibits in the graph structure. It generates a embedding/representation that summarizes the "contextual information" for each input sample, making it easier to separate the correct and misclassified inputs.The contextual attributes extracted by the graph neural network can then be combined with the intrinsic attributes to conduct the better failure-revealing capability estimation (See Sec. 3.4). The contextual attributes extraction process is formally depicted by Algorithm 1.

**Feature Vector Representation (Line 1).** As the target model is to be tested, its feature extraction quality is not guaranteed. Out of this concern, and to make full use of the labeled and unlabeled data, we choose to use a out-of-shelf unsupervised model for feature space construction.

Among the unsupervised learning techniques, the BYOL [Grill et al., 2020] explicitly introduces local continuity constraint into the learned feature space and shows good results on various downstream tasks. Therefore, we train a BYOL model $f_{etr}$ to extract the features from the raw input images. The data used to train the BYOL model includes both labeled and unlabeled data: $(\mathbf{X}_U \cup \mathbf{X}_L)$, and the resulting feature matrix is denoted as $\overline{\mathbf{X}}$. Please note that the feature extractor can be replaced by any other well-trained feature extractor with the local continuity property (*e.g.*, SimCLR [Chen et al., 2020] and MoCo [He et al., 2020]).

**Similarity Graph Construction and Approximation (Line 2).** After the extraction of feature representation, we use the simple distance metric (i.e., cosine) to measure the similarity between any two test instance $\mathbf{x}_i$ and $\mathbf{x}_j$ in $\overline{\mathbf{X}}$: $Dist(i, j) = \mathbf{cosine}(\mathbf{x}_i, \mathbf{x}_j)$. Based on the distance matrix $Dist$, we construct a $k$-NN Graph $\mathcal{G}$, wherein each sample is connected to its top-$k$ most similar samples. The connection is represented by an adjacency matrix $\mathbf{A} \in \mathcal{R}^{N \times N}$, where $N$ is the number of sample in $\overline{\mathbf{X}}$. The entry $\mathbf{A}_{ij}$ equals 1 if node $j$ is within the $k$ nearest neighbors of node $i$, and 0 otherwise. The

edge weight matrix of the similarity graph is denoted as **Edge**, wherein each edge weight in **Edge**, if exists, is inversely proportional to the corresponding distance *Dist*:

$$\mathbf{Edge}_{ij} = \begin{cases} 1/Dist(i,j) & \mathbf{A}_{ij} = 1. \\ 0 & \mathbf{A}_{ij} = 0. \end{cases} \quad i,j \in \{0, \ldots, N-1\}. \tag{2}$$

This means that the connection between two nodes, if exists, is weaker if their proximity is large.

Constructing such a *k*-NN graph is, however, computationally expensive. This is because, calculating the distance between each pair of test instances requires a computational complexity of $O(N^2)$, which is prohibitive to scale up to the current massive unlabeled test instances in real applications. Therefore, we propose an approximation method for *k*-NN graph construction. Our intuition is that, since the target of graph construction is to exploit the failure patterns of the nearby labeled instances for the unlabeled instances, the connections between unlabeled data are less meaningful. Therefore, we propose to only consider the connections among labeled data $\mathbf{X}_L$, and the connections between labeled $\mathbf{X}_L$ to unlabeled data $\mathbf{X}_U$. This approximation reduces the cost from $O(N^2)$ to $O(P^2 + PQ)$, where *P* and *Q* stand for the number of data in $\mathbf{X}_L$ and $\mathbf{X}_U$, respectively. Usually, in the real-world scenario, *P* is much smaller than *Q*, thereby we could obtain a near-linear graph construction algorithm with complexity $O(PQ)$.

**GNN-based representation Learning (Line 3-12).** To apply the GNN algorithm, we first initialize the input node representation matrix $\mathbf{H}^0$ in the similarity graph $\mathcal{G}$ as $\overline{\mathbf{X}}$. Recall that in each GNN layer, the node representations are propagated between neighbors and aggregated together. Thus, we can obtain the representation in the next GNN layer by:

$$\mathbf{H}^{l+1} = \sigma(\tilde{\mathbf{D}}^{-\frac{1}{2}} \tilde{\mathbf{A}} \tilde{\mathbf{D}}^{-\frac{1}{2}} \mathbf{H}^l \mathbf{\Theta}^l), \tag{3}$$

where $\tilde{\mathbf{A}} = \mathbf{Edge} + \mathbf{I}_N$, $\mathbf{I}_N$ is the identity matrix, $\tilde{\mathbf{D}} = \sum_j \tilde{\mathbf{A}}_{i,j}$, $\mathbf{\Theta}^l$ is the trainable weight matrix for the $l^{th}$ layer, $\sigma$ is an activation function and $\mathbf{H}^{l+1}$ is the output representation matrix. The propagation and aggregation are repeated for *M* layers, with the output dimension of the $M_{th}$ layer is 1 (for binary classification purpose).

Then, for any labeled node $\mathbf{x}_e \in \overline{\mathbf{X}}_L$, we could obtain a cross entropy loss between the GNN output $h^M$ and the expected label $y \in \mathbf{Y}_L$ (e.g. misclassified or not): $\mathcal{L}_{ce} = -(y log(h^M) + (1-y)log(1-h^M))$, where $h^M$ denotes probability that $\mathbf{x}_e$ is misclassified. The model is trained via minimizing the loss for some training epochs (we set it as 600 in our experiment). After that, we apply the trained GNN model (except the last layer) on $\mathbf{X}_U$ to obtain the $\mathbf{E}_c$ (line 12). In this way, the correctness of the neighboring samples could be effectively summarized for each node.

### 3.4 Failure-revealing Capability Estimation

To properly combine both the intrinsic attributes $\mathbf{e_m}$ and contextual attributes $\mathbf{e_c}$ for collaborative failure-revealing capability estimation, we formulate the combination function as a simple binary classifier (e.g. a MLP). Specifically, the input to the MLP is a concatenation of $\mathbf{e_m}$ and $\mathbf{e_c}$, and the output is the failure-revealing estimation for an test instance. The final failure-revealing probability is produced by applying a *sigmoid* function $S(t) = \frac{1}{1+e^{-t}}$ on the MLP model's output. We use the labeled instances $(\mathbf{X}_L, \mathbf{Y}_L)$ to train the MLP in a supervised manner, with an objective of minimizing the binary Cross-Entropy loss. After training, the MLP shall re-weight the importance of intrinsic and contextual attributes and make a final decision by assigning a high probability to a test instance if it is likely to reveal a failure.

Finally, we rank the unlabeled test instances in a descending order based on their failure-revealing capability. Under the given budget, we select the top ones to label and test.

## 4  Experiment

### 4.1  Setup

**Datasets.** We evaluate the performance of TestRank on three popular image classification datasets: CIFAR-10 [Krizhevsky et al., 2009], SVHN [Netzer et al., 2011], and STL10 [Coates et al., 2011], as shown in Table 1. More elaboration is shown in the Appendix.

| Dataset | # Class | Size | Official Train/Test/Extra Split | Our Split(TC/DC/HO) | Model Architecture | Model Acc. On HO set(%) (A/B/C) |
|---------|---------|------|----------------------------------|----------------------|---------------------|----------------------------------|
| CIFAR-10 | 10 | 60K | 50K/10K/x | 20K/39K/1K | ResNet-18 | 70.1/66.4/68.3 |
| SVHN | 10 | 630K | 73K/26K/531K | 50K/49K/531K | Wide-ResNet | 94.2/92.5/81.6 |
| STL10 | 10 | 13K | 5K/8K/x | 5K/7.5K/0.5K | ResNet-34 | 54.8/54.0/53.6 |

Table 1: The Dataset and DL Models.

There are mainly two parties involved in the model construction process: the training center and the debugging center. Hence, we manually split the dataset into the training center dataset (see the *TC* column in Table 1) and the debugging center dataset (see the *DC* column in Table 1). To mimic the practical scenario where the unlabeled data is abundant, we move a portion of training data to the debugging center to create this scenario. In the debugging center, we let a set of test data as labeled ones to represent the historical test oracles, and they are 8K/10K/1.5K for CIFAR-10, SVHN, and STL10, respectively, which are used to train the GNN and MLP model. The rest of the data in the debugging center are left unlabeled. Also, we spare a hold-out dataset (see the *HO* column), which is used for evaluating the model accuracy.

**Target DL model (model under test).** As shown in Table 1, we use the popular ResNet and WideResNet architectures as the backbone models [He et al., 2016, Zagoruyko and Komodakis, 2016]. To simulate models of different qualities, for each dataset, we train three DL models with different randomly drawn sub-sets from the training set owned by the training center. For model B and C, the training set are drawn with in-equivalent class weights. After training, we report the accuracy of models on the debugging center's hold-out dataset $T_{HO}$ in Table 1.

**Evaluation metric.** We propose a new evaluation metric for test input prioritization techniques: Test Relative Coverage (*TRC*). *TRC* is defined as the number of detected failures divided by the number of budget or the number of total failures identified by the whole unlabeled test set, whichever is minimal:

$$TRC = \frac{\#Detected\ Failures}{\min(\#Budget, \#Total\ Failures)}. \tag{4}$$

When # budget $\leq$ # total failures, the maximum number of failures can be identified equals to the budget. When # budget $\geq$ # total failures, the maximum number of failures can be detected equals to the total number of failures. Therefore, *TRC* measures how far a test input prioritization technique is to the ideal case.

In practice, under the massive unlabeled data, the performance under a small budget is considered more important than that under a large budget. To provide an insight on the quality of one test input prioritization technique under a small budget, we also provide an ATRC metric: ATRC measures the average TRC values for budget values less than the total failures:

$$ATRC = \frac{1}{N} \sum_{i}^{N-1} TRC_i, \tag{5}$$

where $TRC_i$ stands for the TRC value under budget $b_i, b_i \neq b_j$, and $b_i \leq$ number of total failures.

The proposed metrics enhance the ones used by Feng *et. al.* [Feng et al., 2020] and Byun *et. al.* [Byun et al., 2019]. They use the percentage of detected failures against the percentage of budget (and an APFD [Do and Rothermel, 2006] value derived based on it) for evaluation. Their metric would produce a small value under a small budget, regardless of how good the prioritization technique is. For example, let's assume that there are 10,000 unlabeled data, and 2,000 of them can detect model failures. If the budget is 100, the best percentage of detected failures is 5%, and the worst is 0%. Thus, under their metric, the gap between the best and the worst is only 5%. By contrast, TRC enlarges this gap to 100% to better differentiate the ability of different test input prioritization techniques.

### 4.2 Comparison of TestRank with Baselines

We evaluate *TestRank* against five representative baselines: Random, DeepGini [Feng et al., 2020] (the state-of-the-art), MCP [Shen et al., 2020], DSA [Kim et al., 2019], and Dropout-uncertainty [Byun et al., 2019]. The details of each baseline are illustrated in the Appendix. For the dropout uncertainty method, we run 1000 times inferences with a default dropout rate of 0.5 (the dropout rate is consistent with the one used in [Byun et al., 2019]). For the DSA method, we collect the activation

| Dataset | Model ID | Random | MCP | DSA | Uncertainty | DeepGini | TestRank Contextual-Only | TestRank |
|---------|----------|--------|-----|-----|-------------|----------|--------------------------|----------|
| CIFAR-10 | A | 30.15 | 58.25 | 60.93 | 58.09 | 67.47 | 51.39 | **76.56** |
|          | B | 34.18 | 46.46 | 62.34 | 61.85 | 67.80 | 58.85 | **87.87** |
|          | C | 34.27 | 65.25 | 64.47 | 63.10 | 71.15 | 75.33 | **85.53** |
| SVHN | A | 10.16 | 39.98 | 55.47 | 58.29 | 63.47 | 44.16 | **66.06** |
|      | B | 11.85 | 38.07 | 57.96 | 58.06 | 63.85 | 51.26 | **76.36** |
|      | C | 23.41 | 65.33 | 69.34 | 71.80 | 81.68 | 93.99 | **95.32** |
| STL10 | A | 39.25 | 66.62 | 64.56 | 64.30 | 69.70 | 60.09 | **79.00** |
|       | B | 42.60 | 69.97 | 67.12 | 65.30 | 72.89 | 71.90 | **80.96** |
|       | C | 46.05 | 71.88 | 66.60 | 70.34 | 73.34 | 79.55 | **88.67** |

Table 2: Comparison of *TestRank* with baseline methods with ATRC values (%).

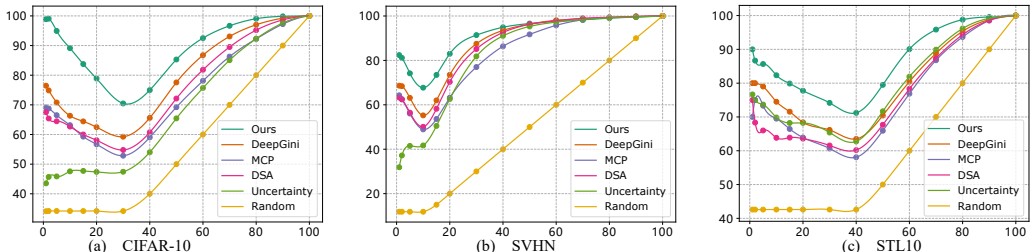

Figure 3: The TRC values against all budgets. X-axis: the budget (%). Y-axis: the TRC value. Note: this figure is generated with model B on each dataset.

traces of the final convolution layer to calculate the surprise score. For our method, we set the number of neighbors for constructing the kNN graph as 100. Also, we use a two-layer GNN with a hidden dimension of 32. More ablation studies are in Section 4.3.

Table 2 compares *TestRank* with baselines using the *ATRC* metric. From this table, we have several observations. First, compared with the baselines, *TestRank* can achieve the highest ATRC values on all evaluated datasets and models. For instance, on CIFAR-10, *TestRank* can achieve 9.09%, 20.07%, 14.38% higher ATRC values than the best baseline DeepGini for model A, B, C, respectively. Therefore, our method can distinguish the failure-revealing capability of the unlabeled test inputs much more accurately. Second, the *testrank-Contextual-Only* column shows the result using only the contextual attributes. We observe that the contextual attributes alone can achieve higher effectiveness than random prioritization. For example, for model A on CIFAR-10, the effectiveness of random prioritization is 30.15% while that of the context-only method is 51.39%. We manually check the distribution of failures of model C and find that many failures are centralized on two classes, where the training data is insufficient. This kind of failure is easily detected by the contextual attributes-based method. Hence, the contextual information is helpful. But still, the context attributes alone are not sufficient. The combination of intrinsic and contextual attributes is essential in achieving high accuracy failure-revealing capability estimation.

To show more detailed results, we present the TRC value against every labeling budget in Figure 3. We observe that the TRC values for most curves decrease in the beginning and then increase. The turning point is when # budget = # total failures. When # budget < # total failures, the TRC values decrease because we rank the test instances according to their failure-revealing probabilities. With the budget increases, the selected test cases have a lower average failure-revealing ability, thus the decreased TRC value. When # budget > # total failures, according to the definition of TRC (see Equation 4), the denominator is fixed. The increase in budget will improve the number of detected failures, and hence the TRC value will increase.

Figure 3 shows that our method consistently outperforms baselines, especially when the budget is small. For example, in Figure 3 (a), *TestRank* improves the prioritization efficiency by around 20% compared to the best baseline when the budget is around 1%. When the budget is rather high (e.g. budget ≥ 80%), the difference between different methods is less obvious because most failures can be selected under the large budget.

| Dataset | Model | TestRank (%) | TestRank w/o approx. (%) | TestRank TargetModel(%) |
|---|---|---|---|---|
| CIFAR-10 | A | 76.56 | 77.77 (+1.21) | 68.84 (-7.71) |
| | B | 87.87 | 87.70 (-0.17) | 81.46 (-6.40) |
| | C | 85.53 | 88.10 (+2.57) | 77.73 (-7.79) |
| SVHN | A | 66.06 | 63.87 (-2.19) | - |
| | B | 76.36 | 82.04 (+5.68) | - |
| | C | 95.32 | 96.62 (+1.30) | - |
| STL10 | A | 79.00 | 80.50 (+1.50) | 67.59 (-11.40) |
| | B | 80.96 | 78.98 (-1.98) | 74.43 (-6.52) |
| | C | 88.67 | 89.32 (+0.65) | 72.43 (-16.23) |
| Average Influence (%) | | | **+0.95** | **-6.23** |

Table 3: The performance (ATRC values) of TestRank under different configurations.

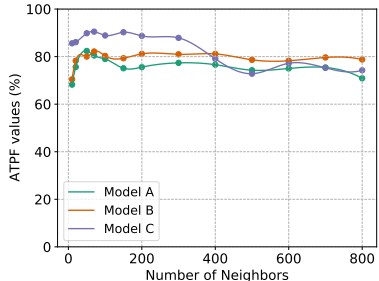

Figure 4: The impact of the number of neighbors $k$ (STL10 dataset).

## 4.3 Impact of TestRank Configurations

**Feature Extractor.** *TestRank* uses an unsupervised BYOL model trained on both labeled and unlabeled data to extract their features. One may wonder if it can be replaced by a supervised model (*e.g.*, the target DL model). To investigate this, we replace the feature extractor in *TestRank* with the front layers (we remove the last few linear layers) of the target DL model. The result is shown in the *TestRank-TargetModel* column in Table 3. Comparing with the original *TestRank*, the average ATRC value on the reported datasets and models reduces by 6.23%, which is significant. As the quality of the given model is to be examined, its feature extraction performance may not be reliable. Also, the dimension of the hidden layer could be huge, making it computationally expensive to calculate similarity values between input samples (This is why we do not report results on the SVHN dataset). In contrast, the separate model is more controllable, enabling it to extract better features for these data. Hence, using a separate feature extractor is necessary.

*k***-NN graph approximation.** To reduce the computation complexity, TestRank uses approximation techniques when constructing the $k$-NN graph (see Section 3.3). The *TestRank-w/o-approx.* column in Table 3 shows the result when we use the original $k$-NN graph without approximation. It indicates that the average influence of the approximation is small (e.g. 0.95%). Therefore, if there is a computation resource limit and the unlabeled test instances are massive, we recommend using the approximation to save computation with negligible performance loss greatly.

**Number of nearest neighbors** $k$**.** When constructing the $k$-NN graph, the number of neighbors $k$ decides the range of the context one node can reach. In previous experiments, the $k$ is set to 100. We enlarge this range to (20 - 800) to study the influence. The result is shown in Figure 4.3. One can observe that the prioritization effectiveness will decrease when $k$ is too small or too large. When $k$ is too small, the context information available to one instance is limited, making it difficult for the GNN to extract valuable contextual attributes. On the other hand, when $k$ is too large, the GNN may grasp irrelevant/noisy information. Still, TestRank can achieve good performance in a wide range of $k$ values. For example, for model A, TestRank is better than the best baseline 69.70% (see Table 2) when $k$ is larger than 20. Hence, selecting the number of nearest neighbors $k$ is relatively flexible.

## 5 Conclusion

We propose *TestRank*, a novel test input prioritization framework for DL models. To estimate a test instance's failure-revealing capability, *TestRank* not only leverages the intrinsic attributes of an input instance obtained from the target DL model, but also extracts the contextual attributes from the DL model's historical inputs and responses. Our empirical results show that *TestRank* outperforms existing solutions significantly. At the same time, this paper considers each test case equally and aims to identify as many failure-revealing test cases as possible. In practice, the impact of each test case could be different. We leave the study of such impact for future work.

## Acknowledgment

This work is supported in part by General Research Fund (GRF) of Hong Kong Research Grants Council (RGC) under Grant No. 14205018 and No. 14205420.

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

# A Appendix

## A.1 Experiment details

Our code is implemented with the open-source PyTorch (under BSD license) [Paszke et al., 2019] and PyG (under MIT license) [Fey and Lenssen, 2019] ML libraries. All of our experiments are performed on a single TITAN V GPU [2].

### A.1.1 Datasets

CIFAR-10 is officially composed of 50,000 training images and 10,000 test images, and it has ten classes of natural images. The Street View House Numbers (SVHN) dataset contains house numbers from Google Street View images. It contains 73,257 training images and 26,032 testing images. Besides, the SVHN dataset also has an extra set of 531,131 images. The STL10 dataset contains ten classes of natural images. In each class, there are 500 training images and 800 test images.

Please note that the datasets we used are open-sourced and available for research purposes [3] [4] [5]. Also, since the data we used is about numbers and the animals, we assume there are no personally identifiable information or offensive content.

### A.1.2 Baselines

Given a DL model and a specific budget, the goal of our method is to select test cases from an unlabeled data pool to discover the failures of the given DL model. We compare our work with the following test input prioritization techniques:

- **DeepGini [Feng et al., 2020]**: DeepGini is the state-of-art test case selection technique. DeepGini ranks unlabeled test cases by a score defined based on the output confidence.

- **MCP [Shen et al., 2020]**: In addition to the output confidence, MCP also considers the balance among different class boundaries of the selected test inputs. Specifically, MCP groups test cases into different clusters, where each cluster stands for a distinct classification boundary, and equally choose low confidence test cases from each cluster.

- **DSA [Byun et al., 2019]**: Byun *et. al.* propose to use the distance-based surprise score (DSA) as a test input prioritization metric, which was initially proposed in [Kim et al., 2019]. The surprise score measures the distance between the test case to the training set. Samples with higher surprise scores are prioritized.

- **Uncertainty [Byun et al., 2019]**: The uncertainty is calculated as the entropy on the averaged output probabilities by running the model multiple times (e.g. $t$ times) with a specific dropout rate.

- **Random:** Test inputs are randomly drawn from all unlabeled samples.

## A.2 Evaluation on More Model Structures

We evaluate our TestRank on two extra architectures: ShuffleNet and MobileNet. The results are shown in Table 4. The model ID indicates the same meaning (e.g., model training with different parts of the training set) as our main paper (see Section 4.1-Target DL model). As we can observe, TestRank can generalize to other models and architectures.

## A.3 Limitation

**Run-time overhead.** We compare TestRank with baseline methods in terms of the run-time overhead in Table 5. However, despite the considerable performance improvement in test input prioritization, TestRank introduces a longer run-time than most baselines. The main reason is due to the calculation of the distances between input features for $k$-NN graph construction. This limitation hinders the application of TestRank in large datasets, and we will solve it in our future work.

---

[2] https://www.nvidia.com/en-us/titan/titan-v/
[3] http://ufldl.stanford.edu/housenumbers/
[4] https://cs.stanford.edu/ acoates/stl10/
[5] https://www.cs.toronto.edu/ kriz/cifar.html

| Dataset | Architecture | ID | Accuracy | Random | MCP | DSA | Uncertainty | DeepGini | Ours |
|---|---|---|---|---|---|---|---|---|---|
| CIFAR-10 | ShuffleNet | A | 83.1 | 19.13 | 57.16 | 56.69 | 37.63 | 64.33 | 66.76 |
| | | B | 79.1 | 23.02 | 56.82 | 59.03 | 37.55 | 65.95 | 76.15 |
| | | C | 75.6 | 24.93 | 60.23 | 62.54 | 51.23 | 69.64 | 79.55 |
| | MobieNet | A | 79.5 | 21.94 | 58.08 | 56.99 | 53.86 | 64.83 | 66.85 |
| | | B | 73.5 | 26.21 | 56.28 | 62.45 | 49.92 | 64.72 | 80.71 |
| | | C | 72.9 | 28.08 | 62.61 | 67.18 | 68.11 | 73.09 | 80.84 |
| SVHN | ShuffleNet | A | 96.88 | 6.11 | 53.58 | 52.32 | 5.12 | 59.18 | 59.77 |
| | | B | 96.56 | 6.97 | 54.76 | 55.95 | 5.84 | 59.74 | 64.32 |
| | | C | 95.68 | 8.55 | 61.90 | 58.34 | 7.01 | 68.62 | 78.03 |
| | MobieNet | A | 96.55 | 7.08 | 53.94 | 55.87 | 12.98 | 62.55 | 61.91 |
| | | B | 95.46 | 8.93 | 55.43 | 52.57 | 14.32 | 62.44 | 68.63 |
| | | C | 94.14 | 11.41 | 58.65 | 71.50 | 15.24 | 69.19 | 81.60 |
| STL10 | ShuffleNet | A | 71.8 | 25.9 | 56.86 | 57.42 | 64.23 | 64.94 | 67.42 |
| | | B | 69.8 | 27.15 | 59.50 | 54.68 | 62.87 | 64.5 | 68.71 |
| | | C | 63.4 | 32.63 | 59.19 | 58.02 | 66.13 | 66.3 | 84.40 |
| | MobieNet | A | 62.8 | 32.28 | 58.40 | 58.65 | 66.69 | 67.6 | 72.22 |
| | | B | 60.8 | 35.71 | 61.48 | 67.62 | 67.75 | 66.52 | 72.73 |
| | | C | 60.6 | 38.7 | 64.16 | 66.58 | 70.59 | 70.01 | 86.46 |

Table 4: Additional comparison of *TestRank* with baseline methods on ATRC values (%).

| Dataset | Random | MCP | DSA | Uncertainty | DeepGini | Ours |
|---|---|---|---|---|---|---|
| CIFAR10 | 7 | 15 | 779 | 103 | 9 | 747 |
| STL10 | 5 | 9 | 627 | 108 | 6 | 142 |
| SVHN | 16 | 33 | 7931 | 1413 | 18 | 1020 |

Table 5: Run-time overhead comparison of *TestRank* with baseline methods (s).

**Failure diversity.** Besides the number of detected failures, failure diversity is another important factor for model debugging. In this work, we have the implicit assumption that the historical labeled data can cover the input distribution, and under such circumstances, TestRank can prioritize unlabeled tests effectively. If, however, the historical test data is severely biased, before prioritizing tests with TestRank, we should analyze the test pool and try to fill this gap first. Otherwise, the detected failures are lack diversity. We shall consider this problem in our future research.

## A.4 Broader Impacts

This work targets building an efficient and effective test input prioritization technique for deep learning models, which can help ensure the security of deep learning models after deployment. Various safety-critical tasks, such as autonomous vehicles, industrial robotics, and medical diagnosis, can benefit from test input prioritization techniques. We consider the scenario in which the unlabeled data is abundant, and a subset of unlabeled data is selected for labeling and testing. While building the unlabeled data by collecting them from the Internet or other sources, it may have a chance to include some unauthorized or private data. Also, the collected unlabeled data could be biased, resulting in the testing being incomplete. To avoid such cases, we should always guarantee that the collected data will not violate any privacy or rights and ensure that the collected data cover instances as many as possible.

