# OpenReview forum: "TestRank: Bringing Order into Unlabeled Test Instances for Deep Learning Tasks"
_NeurIPS.cc/2021/Conference — NeurIPS 2021 Poster_

### Official Review · Reviewer_7iHb · 2021-07-12

**Rating:** 6
**Confidence:** 4

**Summary:**

- Given a dataset of labeled and unlabeled examples, authors build a
  data point prioritization technique, which helps choose data points
  to be labeled next, with a goal of revealing failures as soon as
  possible. The technique should help prioritize labelling instances
  which currently trained model would misclassify.
- They use intrinsic and contextual attributes to build a model
  performing data point prioritization. Intrinsic attributes are those
  that are obtained by observing outputs of the deep learning model
  for a given input. Contextual capture information about similarity
  to the other instances -- if a given instance is similar to
  misclassified instances, that would be a strong signal that the
  instance is problematic.
- For intrinsic attributes, they use the output layer logits (before
  the softmax activation) of the DL model to construct a vector to be
  used for a final model. Second, they use a graph neural network to
  create contextual representation vector out of both labeled and
  unlabeled data. The aforementioned vectors are input to an
  multi-layer perceptron (MLP) model. The output of the MLP is a
  probability that a given input would be misclassified by the DL
  model.
- They evaluate their model against other data point prioritization
  techniques measuring number of failure revealing instances for a
  given labelling budget.

**Limitations And Societal Impact:**

Regarding the limitations, authors recognize that not each data
instance prioritized is of the same importance, which they did not
take into consideration. I think another important point to mention is
that while the technique learns to prioritize instances for a single
model, it is unclear if these instances are in any way special, or
they are just specific to the model trained. This is important if our
goal is to produce the best quality dataset. Alternatively, if the
idea is to keep retraining the model with the additional data it would
be good to evaluate how model benefits when augmented with newly
labeled data points.


**Main Review:**

- I find the problem being solved an important one as manual labelling
  costs can become prohibitive. The problem being solved should help
  maximize the gains from manual labelling efforts providing a better
  quality model with less data overall.

- Considering that the overall goal is to maximize the quality of the
  data we label and generality of the model we train using that data,
  I wonder if data point prioritization techniques should be run on a
  single model or multiple models trained. We only know that the
  instances prioritized are troublesome for the current model, but are
  they really special? What if we trained another model with the same
  data we have, would we end up prioritizing different inputs?

- I would like to see evaluation on more models. More precisely, I
  would like to see for a single dataset having couple of model
  architectures, and for each architecture K models trained. Then,
  evaluate data point prioritization techniques on K models for each
  architecture. This is to gain more confidence that the performance
  gains generalize across models and architectures.

- I find it slightly difficult to read the paper. For example, on line
  256 it is written: "8K/10K/1.5K", but it is not written what the
  splits are for. Then, the following sentence sounds incomplete:
  "when ..., the maximum number of failures can be identified by a
  test prioritization technique equals to the budget". Also, following
  one: "which exploits the relational inductive-bias that exhibits in
  the graph structure". Also, following sentence: "we assume that
  there are 10,000 unlabeled data, and 2,000 of them can detect model
  failures", should probably start with: "let's assume".

- I see many mentions of "test prioritization" in the
  paper. Personally, I would find "data point prioritization", or
  "instance prioritization" a more natural name than the "test
  prioritization". The reason is that in software engineering, in the
  case of test case prioritization, there is already an oracle
  predefined for the tests being prioritized, while in this case we
  are about to create the oracle (label a prioritized instance). There
  are also mentions of "test instance prioritization" and "test input
  prioritization" which sound more natural. What might help paper read
  better is to use the naming consistently.

- Figure 3. It's hard to distinguish colors of different approaches in
  the legend.

**Time Spent Reviewing:**

4

---

> ### Author Response · Authors · 2021-08-10
> **Response to Reviewer 7iHb:**
>
>
> **Q1: Considering that the overall goal is to maximize the quality of the data we label and generality of the model we train using that data, I wonder if data point prioritization techniques should be run on a single model or multiple models trained. We only know that the instances prioritized are troublesome for the current model, but are they really special? What if we trained another model with the same data we have, would we end up prioritizing different inputs?**
>
> A: If we train another model with the same data we have, we will prioritize different test inputs because our test input prioritization technique is model-specific. In practice, we believe some of the prioritized test inputs are only special to the model, while the others are special to the task. Both kinds of failures (e.g., failure cases to one specific model and failure cases to multiple models) are valuable for model improvement. For example, if we have two models VGG and ResNet, which are good at capturing different parts of the training dataset, and we identified their corresponding failure cases. The intersection of their failures would be cases specific to the task and is helpful to improve both models. Also, the failures specific to one model are also valuable to improve the specific model. Note that, TestRank aims to find all failures for a target model. If one wants to find the common failures for multiple models, he/she can apply our TestRank on multiple models and take the intersection among them.
>
> **Q2: I would like to see evaluation on more models.**
>
> A: We evaluate our TestRank on two extra architectures: ShuffleNet and MobileNet. The results are shown in the following table. The model ID indicates the same meaning (e.g., model training with different parts of the training set) as that in our paper (see Section 4.1-Target DL model). As we can observe, TestRank can generalize to other models and architectures. Here we only show the results on the SVHN dataset. We shall evaluate the two architectures on CIFAR10 and STL10 datasets in the updated version.
>
> Table: Comparison of TextRank with baseline methods with ATRC values (%).
>
> |        Architecture           |     ID    |     Accuracy    |     Random    |      MCP     |      DSA     |     Uncertainty    |     DeepGini    |     TestRank    |
> |:-----------------:|:---------:|:---------------:|:-------------:|:------------:|:------------:|:------------------:|:---------------:|:---------------:|
> |     ShuffleNet    |      A    |       96.88     |      6.11     |     53.58    |       55.32      |         5.12       |       59.18     |       63.09     |
> |                   |      B    |       96.56     |      6.97     |     54.76    |       55.95      |         5.84       |       59.74     |       63.34     |
> |                   |      C    |       95.68     |      8.55     |     61.90    |       58.34      |         7.01       |       68.62     |       73.61     |
> |      MobieNet     |      A    |       96.55     |      7.08     |     53.94    |     55.87    |        12.98       |       62.55     |       60.13     |
> |                   |      B    |       95.46     |      8.93     |     55.43    |     52.57    |        14.32       |       62.44     |       66.9      |
> |                   |      C    |       94.14     |      11.41    |     58.65    |     71.50    |        15.24       |       69.19     |       73.74     |
>
>
> **Q3: I find it slightly difficult to read the paper.**
>
> A: First, as indicated in our setup session, "8K/10K/1.5K" stands for the number of previously tested samples for dataset CIFAR10/SVHN/STL10, respectively (line 256). For the later grammar issues, we shall fix them in the updated version.
>
>
> **Q4: What might help paper read better is to use the naming consistently.**
>
> A: We shall make it consistent in our updated version.
>
> **Q5: Figure 3. It's hard to distinguish colors of different approaches in the legend.**
>
> A: We shall make it more clear in our updated version.

---

### Official Review · Reviewer_d2bi · 2021-07-16

**Rating:** 6
**Confidence:** 3

**Summary:**

This paper proposes a new test sample prioritization strategy to test deep learning systems. The objective is to maximize the number of failure-inducing samples selected from unlabeled samples. The authors propose to leverage both the samples' intrinsic properties (from the model's prediction behavior on them) and contextual properties (from other labeled samples close to them) as the criteria to select samples. By learning another binary classifier that takes the two properties as input, TestRank produces probabilities to rank the test samples. Empirical results demonstrate that TestRank outperforms existing test prioritization techniques.

**Limitations And Societal Impact:**

Yes

**Main Review:**

####Stregths:
- Clear description of the technique.
- Using contextual features generated by KNN+GNN seems novel.

####Weakness:
- Lack of justification on why samples close to incorrectly-labeled samples should be prioritized.
- Unclear why GNN is needed to extract contextual property -- how it compares to simple alternatives.

####Comments:

Overall, the paper is well-written. The motivation and the methodology are clearly described and very easy to understand. Having said that, my main concern is that I am not convinced by the criterion that the authors choose to prioritize test samples. I understand the authors' motivation is that test samples close to known error-inducing samples should be prioritized, as these samples are more likely "failure-revealing ." However, this assumption raises two questions:

First, if failure-revealing is the only criterion, numerous adversarial testing papers have been shown very effective in constructing error-inducing samples. The generated samples can also be far away from the decision boundary. Besides, the generated samples can be regularized to match real-world image distribution (using GANs or hard constraints). Given that adversarial testing can generate an unbounded number of real-world failure-revealing test samples, I do not see the clear use-case or unique advantage of test prioritization using only existing (limited number of) samples.

Second, I think the ultimate goal of testing is to reveal model violations of some abstract properties. For example, for computer vision models, we want to test whether the model is robust to a certain class of transformation (lighting conditions), labels, background, object poses, etc. Therefore, the testing should uncover diverse samples that can reveal different property violations instead of keeping generating samples violating the same properties. The idea is similar to traditional software testing, where we are interested in constructing tests that trigger different code paths. So why prioritize those samples that we already know the model is highly likely to fail?

As of methodology, I do not find the justification for using GNN (in Section 3.3) convincing. I understand that measuring the proximity of images is challenging, but why not directly leverage the target model, e.g., using its last convolution layer (often used to produce image representations), to generate the representation and compute the distance? Since the contextual property identifies the close samples from the target model's perspective, it seems natural to use its internal representation of the inputs to compute distances. This also saves the scalability issue of GNNs when working on a large number of nodes.

###Minor:

Sometimes "TestRank" is named as "TextRank". Are they indicating the same thing?

Does training the binary classifier (for failure-revealing) also update the weight parameters of the GNN? It is interesting to study which strategy works better - updating GNNs or keeping them frozen.

Approximate nearest neighbor (ANN) might be a very good alternative to KNN as the authors are approximating the computation anyway (ignoring unlabeled samples). ANN is much more efficient than KNN.

"et. al."->"et al."

**Time Spent Reviewing:**

8

---

> ### Author Response · Authors · 2021-08-10
> **Response to Reviwer d2bi:**
>
> **Q1: Given that adversarial testing can generate an unbounded number of real-world failure-revealing test samples, I do not see the clear use-case or unique advantage of test prioritization using only existing (limited number of) samples**.
>
> A: We distinguish testing with clean samples from testing with adversarial ones since they belong to two different development stages in the deep learning model development life cycle: the functional development stage and the robustness enhancement stage.  In the functional development stage, the developers need to conduct functional testing. The testing inputs are natural and clean ones that a benign user would input, and they are used to find the DL model's functional bugs on the specified task. While in the robustness enhancement stage, the developers need to generate test samples with different attack methods to discover system vulnerabilities.  Both testings are necessary to produce a good model, and functional testing usually comes before security testing: the security testing is less meaningful if the model architecture and the corresponding parameters are not certain yet. In this paper, we focus on the functional testing part, and that's why we use the benign inputs for testing.
>
> **Q2: So why prioritize those samples that we already know the model is highly likely to fail?**
>
> A: We agree with the reviewer that failure diversities are important for test and debug. As mentioned in our paper, we assume that the labeled samples have already fulfilled the test diversity requirement. Given this, discovering more failure samples from the unlabeled pool is still of great value.  Different from traditional software testing, where finding one failure is often sufficient for fixing the corresponding bug, a single failure instance is not statistically meaningful in the deep learning setting. We need to uncover as many related failures as possible to discover their common failure patterns.
>
> **Q3: As of methodology, I do not find the justification for using GNN (in Section 3.3) convincing. I understand that measuring the proximity of images is challenging, but why not directly leverage the target model, e.g., using its last convolution layer (often used to produce image representations), to generate the representation and compute the distance? Since the contextual property identifies the close samples from the target model's perspective, it seems natural to use its internal representation of the inputs to compute distances. This also saves the scalability issue of GNNs when working on a large number of nodes.**
>
> A: First, the GNN is used to summarize the classification correctness of the neighboring labeled samples, as the GNN is good at capturing the local properties in a high-dimensional manifold. To apply the GNN algorithm, we must measure the distances between nodes to construct the graph. Measuring the distances using the feature extracted either from the target model or from a separate model are both applicable. We experimentally show that if we use the features extracted from the target model for graph construction, the failure recall rate is less effective than the current Rank (See Table 3 for more details). The reason behind this is that, given a model under test, its feature extraction ability may not be reliable and can be misleading, due to overfitting or other training defects. Finally, for the scalability issue, except for the technique we proposed for our task, there are many existing works that could be used to make the GNN more scalable [1,2].
>
> [1] Chiang, W. L., Liu, X., Si, S., Li, Y., Bengio, S., & Hsieh, C. J. (2019, July). Cluster-gcn: An efficient algorithm for training deep and large graph convolutional networks. In Proceedings of the 25th ACM SIGKDD International Conference on Knowledge Discovery & Data Mining (pp. 257-266).
> [2] Zeng, H., Zhou, H., Srivastava, A., Kannan, R., & Prasanna, V. (2019). Graphsaint: Graph sampling based inductive learning method. arXiv preprint arXiv:1907.04931.
>
> **Q4: Sometimes "TestRank" is named as "TextRank". Are they indicating the same thing?**
>
> A: Yes, it is a typo. we will revise it accordingly.
>
> **Q5: Does training the binary classifier (for failure-revealing) also update the weight parameters of the GNN?**
>
> We make the GNN frozen when training the binary classifier. We have tried to update the GNN with the classifer, but the results are much worse.  We believe this is because updating the GNN with the classifier would make the contextual features lose their original meaning, i.e., classification correctness of neighboring nodes.
>
>
> **Q6: Approximate nearest neighbor (ANN) might be a very good alternative to KNN as the authors are approximating the computation anyway (ignoring unlabeled samples). ANN is much more efficient than KNN.**
>
> A: The intuition of our approximation method is that the target of graph construction is to exploit the failure patterns of the nearby labeled instances for the unlabeled instances, and the connections between unlabeled data are less meaningful. Therefore, when searching for the nearest neighbors, we can omit the distance calculation between some nodes (e.g., an unlabeled node to another unlabeled node) to reduce the candidate neighbor nodes. In addition to this, for the rest candidate neighbor nodes, we can apply ANN methods to further reduce the computation. To sum up, we can use the existing ANN methods in conjunction with our approximation method to improve the computation speed.
>
> **Q7: "et. al."->"et al."**
> A: We will revise it accordingly.

---

> > ### Comment · Reviewer_d2bi · 2021-09-01
> > **Thanks for the response**
> >
> > I appreciate the authors' effort in addressing my questions. I have increased my score to 6. However, I would hope the authors could address my follow-up questions in their next version.
> >
> > Q1. I hope to see more discussions on the rationale of having the functional development stage and the robustness enhancement stage. Are there any known references or well-accepted specifications that evaluating ML model should have such stages? I think there is no clear boundary between the two, as they are all talking about correctness or some properties that the model must meet, e.g., classifying dog images meaning it must correctly predict dog images with different breeds, poses, background, natural transformation, etc. These images can appear naturally in the functional test set and can also be created/generated to evaluate the model's robustness.
> >
> > Q2. Could you show the failure patterns uncovered by TestRank for different models, given that you argue finding similar failures is statistically important? Also, I think finding diverse failures is complementary to finding multiple instances of one type of failure. As a minor point, the authors should at least discuss the early ML testing works, such as DeepXplore and DeepTest, about the coverage metric used to evaluate the test samples.
> >
> > Q3. Please add the discussion to the next version. If possible, please report the runtime overhead and compare it to other baselines.

---

> > > ### Author Response · Authors · 2021-09-02
> > > **Thank you for raising the score**
> > >
> > > We thank the reviewer for the comments, and we address the reviewer's concerns as follows:
> > >
> > > **Q1: I hope to see more discussions on the rationale of having the functional development stage and the robustness enhancement stage. Are there any known references or well-accepted specifications that evaluating ML model should have such stages? I think there is no clear boundary between the two, as they are all talking about correctness or some properties that the model must meet, e.g., classifying dog images meaning it must correctly predict dog images with different breeds, poses, background, natural transformation, etc. These images can appear naturally in the functional test set and can also be created/generated to evaluate the model's robustness.**
> > >
> > > The rationale for distinguishing the functional and robustness/security tests originates from better quality assurance for real-world software systems (see [1] for more details). In this context, the functional test ensures the system follows the functional specification, while the security test is to ensure the system meets the security requirements in the presence of adversaries. The same principle is applied to machine learning systems. Zhang et al. [2] summarize the related paper on machine learning testing, wherein the functional correctness and robustness belong to two different testing properties. Functional testing often comes before security testing because we must have a working model at first.
> > >
> > > At the same time, we agree with the reviewer that robustness is an ambiguous term without an exact definition, and we could also use natural images for "robustness testing." As a result, we do not emphasize the very purpose of test prioritization in our paper. The problem is formulated as prioritizing natural test instances (i.e., instead of generating tests, e.g., adversarial examples) based on their failure-revealing capabilities.
> > >
> > > We do not consider test generation in this paper because, practically speaking, there are usually abundant natural samples for tests in many fields and the key constraints lie in the labeling budget.
> > >
> > > [1] Dashti, Mohammad Torabi, and David Basin. "Security testing beyond functional tests." International Symposium on Engineering Secure Software and Systems. Springer, Cham, 2016.
> > >
> > > [2] Zhang, Jie M., et al. "Machine learning testing: Survey, landscapes and horizons." IEEE Transactions on Software Engineering (2020)."
> > >
> > > **Q2: Could you show the failure patterns uncovered by TestRank for different models, given that you argue finding similar failures is statistically important?**
> > >
> > > One known example of a DL-based classifier bug is that the classifier uses the snow features  (instead of the features of the wolf itself) to classify the snow wolf. Consequently, if the snow wolf appears in a different environment, it will be misclassified. If we have just one failure sample, it would be challenging to reason such a bug. However, it would be relatively simple to see the difference between correct and failed samples and resolve the bug if we have many failure samples. Moreover, even if we do not reason the bug and simply rely on model retraining to improve the model, retraining with a single previously failed sample can hardly benefit. In contrast, retraining with multiple previously failed samples has a higher probability of fixing the bug.
> > >
> > > **Q3: Also, I think finding diverse failures is complementary to finding multiple instances of one type of failure.**
> > >
> > > We fully agree with the reviewer on this comment. This is a fundamental problem related to defining an accurate yet computationally efficient data-driven coverage for ML systems, and we leave it for future work.
> > >
> > > **Q4: As a minor point, the authors should at least discuss the early ML testing works, such as DeepXplore and DeepTest, about the coverage metric used to evaluate the test samples ... If possible, please report the runtime overhead and compare it to other baselines.**
> > >
> > > We thank the reviewer for the suggestions. We shall add the corresponding discussions on these prior works and report the runtime overhead in the updated version.

---

### Official Review · Reviewer_i1xT · 2021-07-19

**Rating:** 6
**Confidence:** 2

**Summary:**

This paper introduce a way to test the generalization performance of DNNs using labeled together with unlabeld datasets via less labeling efforts. Authors proposed  TestRank -- an integrated approach that
(1) adopts an semi-supervised learning algorithm to "label" every unlabeled sample using a vector of discriminative features using GNN and label data.
(2) retrieves features of every unlabeled sample from the DNN model under testing, and
(3) trains an MLP using both GNN-predicted features and DNN-extracted features as inputs to predict failures of the DNN model as targets.
Some evaluation has been done using three image classification datasets.

**Ethics Review Area:**

["I don’t know"]

**Limitations And Societal Impact:**

This paper introduces a way to test the generalization performance of DNNs incorporating unlabeled datasets (validation-free information).

**Main Review:**

Testing generalization performance of DNNs with unlabeled datasets is an important problem of deep learning. I am wondering whether the proposed problem formulation "test prioritization" really makes sense in ML societies (I noted the most of cited papers were from the software engineering domain).

Actually, there are a series of challenges and works on predicting the generalization gap using unlabeled datasets or validation-free information, such as NIPS 2020 Challenge "Predicting Generalization Gap" . Furthermore, there are also some papers in this domain such as "In Search of Robust Measures of Generalization" at NIPS 2020, and "Predicting the Generalization Gap in Deep Networks with Margin Distributions" at ICLR 2019.

Two simple yet effective baseline to compare with TestRank are as follows.
(1) Using random data augmentation to generate pseudo testing samples using the training dataset with labels preserved, and predicting the generalization gap using the pseudo testing samples and the training samples. The gap of classification error between original samples and augmented samples could simply predict the generalization performance.

(2) Using random data augmentation to generate an additional set of unlabeled dataset using the original unlabeled datasets, and verifying whether the outputs of the DNN model based on every pair of images (the original image and the augmented image) are consistent. Such inconsistency should also characterize the generalization gap.


**Time Spent Reviewing:**

2hrs

---

> ### Author Response · Authors · 2021-08-10
> **Test input prioritization v.s.  generalization gap prediction.**
>
> We thank the reviewer for the comments, but we would like to point out that predicting generalization performance and test input prioritization are **two different problems**.
>
> First, the objective of the former one is to predict the performance gap of a trained deep learning model between the training dataset and the unseen test dataset having the same distribution as the training dataset. In contrast, in the test input prioritization scenario, the training data and the unlabeled test data may not follow the same distribution, e.g., due to practical issues, the training data can be biased and is not representative of the task. Second, test input prioritization can provide detailed failure information at the instance level, while generalization gap prediction often cannot. Also, test input prioritization aims to conduct efficient testing with less labeled data, while efficiency is not the primary concern of generalization gap prediction. Due to the above, the two baseline methods mentioned by the reviewer do not apply to the problem investigated in our paper.

---

> > ### Comment · Reviewer_i1xT · 2021-08-31
> > **I agree but...**
> >
> > Many thanks for the responses. Now I am clear about the difference between generalization gap prediction and test input prioritization. However, for instance level, we can still do the same thing as the method mentioned in  NIPS 2020 Challenge "Predicting Generalization Gap". For any unlabeled sample, we can use the testing DNN to obtain its (soft) prediction result, i.e., a logit. Then, we use random augmentation to generate various augmented copies of such sample with different views. We can pass these augmented samples through the testing DNN and obtain their logits. We could compare these logits and measure the average distance between these logits. When the distance is high, we could doubt that the testing DNN might not work well on such sample (even though we don't know the ground truth label). At least, I guess it would be a simple yet effective baseline for comparisons.

---

> > > ### Author Response · Authors · 2021-09-01
> > > **Thanks for the suggestions**
> > >
> > > We thank the reviewer for proposing an interesting baseline method for test input prioritization.
> > >
> > > We implement a simple method based on this idea (denoted as **RandomAug**) and compare it with TestRank. To be specific, we use random rotation to generate 10 augmented copies for each image and measure the output variations to determine their failure-revealing probabilities. If the output logits of an image and its copies have higher variation, this image is considered to have better error-revealing probability. We compare RandomAug with TestRank on CIFAR-10, and the results are shown in the following table:
> > >
> > > |  Ranking Method |   A   |   B   |   C   |
> > > |:---------:|:-----:|:-----:|:-----:|
> > > | RandomAug | 17.98 | 34.69 | 34.59 |
> > > |  TestRank | 76.56 | 87.87 | 85.53 |
> > >
> > > Table: Comparison of *TestRank* with *RandomAug* on ATRC values (%).
> > >
> > > The simple augmentation method that we use may not fully realize the potential of the proposed idea, but the huge performance gap to TestRank at least shows that such a baseline method needs significant improvements to perform well.

---

> > > > ### Comment · Reviewer_i1xT · 2021-09-01
> > > > **Thanks**
> > > >
> > > > Thanks for your reply. I have upgraded the score.

---

### Decision · Program_Chairs · 2021-09-27

**Decision:**

Accept (Poster)

**Comment:**

The reviewers appreciated both the interesting problem of prioritizing unlabelled data points as well as the graph-based similarity technique to predict failure-revealing capabilities of data points. There were some concerns regarding baselines, additional evaluation tasks, and motivation of design choices, but the responses clarified many of these concerns. It would be great if authors can incorporate the additional clarifications and experimental results in the final version.